# Characterization of Antenna Radiation Pattern and Penetration Depth in Ground Penetrating Radar Field Missions

**Pavel Morozov [1,2], Fedor Morozov [2], Maxim Lazarev [3], Leonid Bogolyubov [4] and Alexei Popov [1,*]**

1 Pushkov Institute of Terrestrial Magnetism, Ionosphere and Radio Wave Propagation (IZMIRAN), Troitsk, 108840 Moscow, Russia; pmoroz5@yandex.ru

2 JSC Company VNIISMI, 127566 Moscow, Russia; fmorozov92@mail.ru

3 Faculty of Fundamental Physics, Moscow Pedagogical State University (MPGU), 118881 Moscow, Russia; m.a.x.i.m.2000@mail.ru

4 Faculty of Managemrnt, Bauman Moscow State Technical University (MSTU), 105005 Moscow, Russia; l.e.o.n.2002@mail.ru

* Correspondence: popov@izmiran.ru; Tel.: +7-910-400-91-07

**Abstract:** This article discusses the methods and results of assessing the angular resolution and sounding depth of enhanced-power ground penetration radars obtained during archaeological and geographical expeditionary works in various natural areas. Elongated local objects were used as test objects to evaluate the horizontal radiation pattern of the Loza–V georadar in the upper- and lower-half spaces. The depth of operation of the Loza–N low-frequency radar was estimated during a geophysical study of a unique natural object in the Siberian taiga. The variability of the GPR antenna radiation patterns in different materials (air, dry, or wet soils) confirms the necessity of quantitative measurements with controlled electrophysical parameters.

**Keywords:** impulse subsurface radar; radiation pattern; archaeological and natural objects

## 1. Introduction

Ground penetrating radar (GPR, georadar) [1–3] is widely used in archeology, geology, civil engineering, and security issues [4–6]. The problem of GPR radiation directivity plays an important role in planning the measurement paths in rural and urban conditions. On the one hand, a wide radiation pattern increases the field of view of the device and makes it possible to estimate the permittivity of the subsurface medium from the attenuation of the signal by separating the transmitter and receiver antennas [1] (other proposed approaches require an improvised test platform [7,8] or double passing the same path with different antennas [9]). On the other hand, especially when working in an urban environment, side reflections make it difficult to detect and identify objects of interest. Of special interest are the long resistively loaded antennas of low-frequency georadars used in geophysics and geological exploration. While the high-frequency antennas can be tested in an electromagnetic anechoic chamber, the performance of long low-frequency GPR antennas can be studied only during practical fieldwork.

In this article, we describe the directional and penetration characteristics of powerful GPR of the Loza series ("deep penetration radar", DPR) [9–11]. All of them have antennas made as dipoles of different lengths, with the resistive loading roughly following Wu–King's law [12] and minimizing "ringing" at the antenna's central frequency. Although the radiating and receiving properties of GPR antennas have been studied in a number of theoretical and experimental works (see [13–20]), georadar missions provide a good opportunity to assess their performance under realistic operating conditions.

In the first part of this paper, we present the results of theoretical and experimental studies of the GPR antenna radiation patterns in the subsurface medium and in the upper

hemisphere (the former characteristic is necessary for the interpretation of georadar B-scans [1], whereas the latter is useful when working in urban conditions, with multiple interfering aerial reflections). These methodological works have been performed by the research team of the Pushkov Institute of Terrestrial Magnetism, Ionosphere, and Radio Wave Propagation (IZMIRAN) and JSC "Company VNIISMI" during a joint expedition organized by the Institute of Archeology of Crimea of the Russian Academy of Sciences in July, 2022. Among the studied archeological objects, the ancient Jewish cemetery in the vicinity of Kerch city, the remains of an ancient water pipe near the village Geroevka (El Tiygen), and the destroyed Venetian fortress in Tikhaya Balka should be mentioned. The results of the field campaign were reported at a scientific-practical conference [21] in December 2022. Along with archaeological research, the features of the local relief and subsurface artifacts were used in order to evaluate the technical characteristics of the Loza–V GPR [10]. First of all, we were interested in estimating the radiation pattern of the resistively loaded dipole antennas used in this enhanced georadar power. This important parameter has been studied in a number of theoretical and experimental works, e.g., see [14,22–25]. However, taking into account the construction, size, and potential of our DPR antennas, designed for geophysical and industrial applications, full-scale field measurements were invaluable.

The second part of the presented experimental material was obtained during the expedition organized in September 2022 by the Russian Geographic Society, the newspaper "Komsomolskaya Pravda" and the TV channel "Russia1". The object of the study was the Patomsky crater, an unusual geological structure in the Siberian taiga discovered in 1949 in the Irkutsk region [26]. This rare object (a stone ring 80 m in diameter with a central cone composed of large limestone fragments) attracted the attention of researchers, and various hypotheses of its origin were put forward. Our GPR cross sections, performed with Loza–N GPR [11], were aimed at choosing between the most probable volcanic model [27] and the early meteorite hypothesis [28]. From a technical point of view, this experiment confirmed the possibility of GPR sounding the first hundreds of meters of a subsurface medium. Geophysics revealed a fundamental difference between deep GPR echoes from a massive stone ground and from a probable water-filled volcanic crater.

## 2. Materials and Methods

Experimental studies of the directional pattern of dipole antennas were carried out using a GPR of the Loza series [10,11]. The main design features of the device were outlined in [29]. In this "deep penetration radar" (DPR) construction, everything was performed to increase its depth sounding capabilities (high voltage EM pulse and long resistively loaded antennas). The design features made it possible to use DPR in the study of subsurface structures at depths of up to 100–150 m in "heavy" low-resistivity soils and up to 200–300 m in high-resistivity rocks.

The fundamental features of the Loza GPR series design are follows:

- Ultra-high power: The peak power of the transmitted EM pulse was brought up to a practical limit, determined only by the insulating properties of the environment, by means of a high-voltage discharger supplying a probing pulse of voltage from 5 to 21 kV to the antenna.
- Concentration of signal energy in the low-frequency part of the spectrum: In order to achieve maximum depths, the maximum energy of the probing pulse is shifted to the lowest frequencies, within the 1–50 MHz frequency band of the receiver determined by the length of the transmitter antenna. The medium-frequency version Loza–V, operating in the frequency band of 50–300 MHz, is equipped with 100, 200, and 300 MHz dipole antennas (Figure 1). The Loza–N DPR is supplied with 50 MHz (3 m long), 25 MHz (6 m), 15 MHz (10 m), and 10 MHz (15 m) transmitter and receiver half-wave dipole antennas (Figure 2). All antennas are designed with a resistive load that gradually grow towards the ends of the dipole [12].

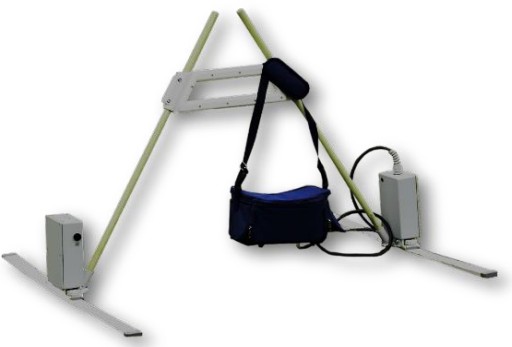

**Figure 1.** Loza–V GPR with 200 MHz (0.75 m long) antennas.

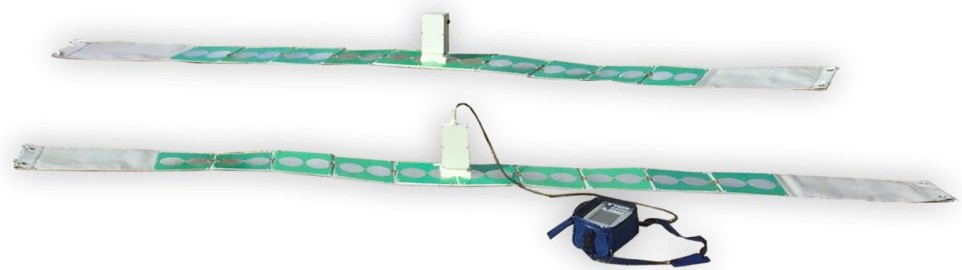

**Figure 2.** Loza–N GPR with 50 MHz (3 m long) antennas.

-   Large dynamic range of the reflected signal registration: The peak power of the transmitter is brought to a physical limit and is limited only by the electrical breakdown of the environment (ground and air). The use of ultra-powerful transmitters and low-frequency antennas provides a dynamic range of reflected signals of more than 120 dB, which allows one to work in environments with high conductivity, such as loam or wet clay. The registration system allows one to digitize the signal in the entire dynamic range without changing the GPR settings. The resistive damping makes it possible to obtain probing pulses of a practically non-oscillating nature [14,22–24], see Figure 3.

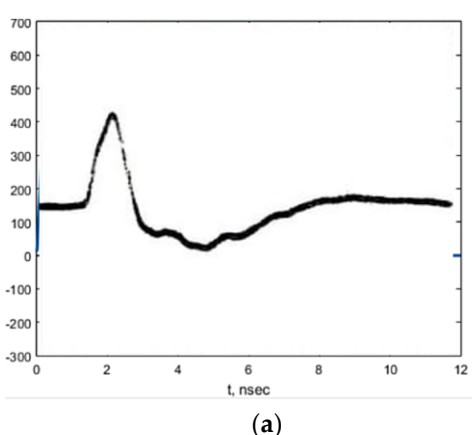

(**a**)

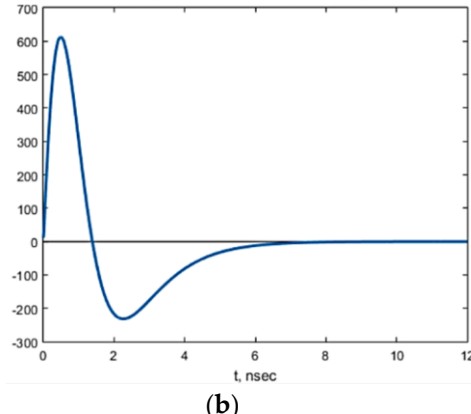

(**b**)

**Figure 3.** (**a**) Measured pulse in a resistively loaded dipole antenna (negative of the original screen [14], a.u.), and (**b**) GPR pulse model [11].

The efficiency of GPR sounding (first of all, its depth) depends on the parameters of the environment. The radiation patterns of the antennas are largely determined by the electrical properties of the ground on which the antennas are placed. When the GPR antenna (linear

electric dipole) is placed tightly at the interface between two media (ground surface), a radiation pattern is formed, essentially oriented towards the medium with higher dielectric permittivity $\varepsilon_r = n^2$ (or soil refractive index) $n = \sqrt{\varepsilon_r}$ [30,31]. Approximately $\varepsilon_r = n^2$ times more radiation is emitted towards the lower hemisphere (into the ground) than upwards (into the air).

Figure 4a,b illustrate the effect of the interface on the harmonic and pulsed EM radiation of an idealized infinite-line antenna into the subsurface medium. In both cases, a noticeable maximum of the radiation pattern in the far zone is observed in the directions corresponding to the angle of total internal reflection $\Theta = \arcsin\frac{1}{n}$, cf. [13,18,32]. These estimates allow one to assess the GPR radiation pattern in the transversal "H plane". A similar but smoother pattern, without "horns", demonstrates the popular GSSI bow-tie antenna [20]. In both cases, directivity is rather poor.

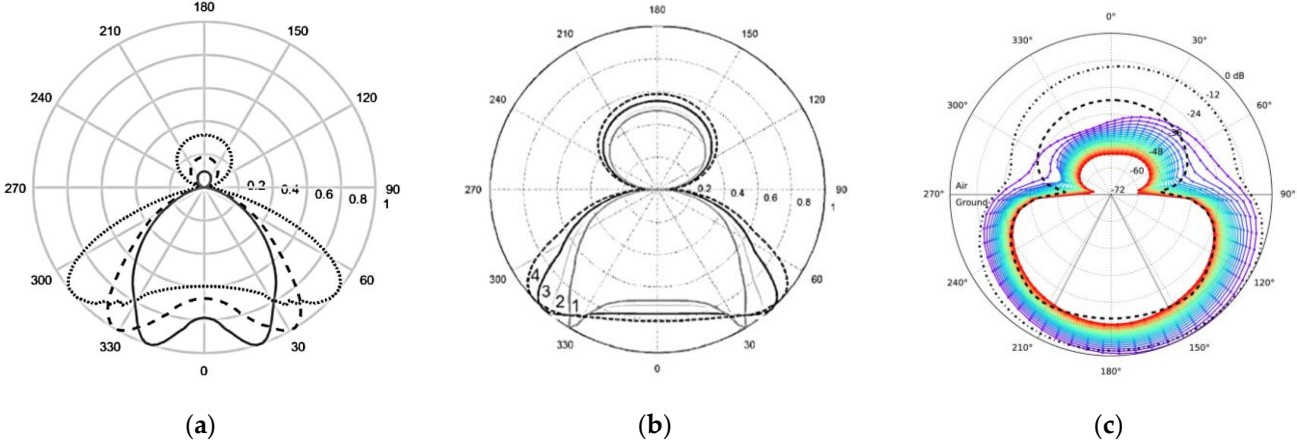

(**a**)　　　　　(**b**)　　　　　(**c**)

**Figure 4.** (**a**) Radiation pattern of an infinite line vibrator on the ground–air boundary [30]. Harmonic radiation and different values of the ground permittivity $\varepsilon_r$ = 2 ($\cdots$); 4 (– – –); 9 (——). (**b**) Peak radiation pattern of an impulse line antenna [31] for $\varepsilon_r$ = 4 and a one-period sinusoidal current pulse form. Different distances between the transmitter dipole and point receiver: (1) 200 m, (2) 12 m, (3) 4 m, and (4) 2 m. (**c**) Total energy radiation patterns of a 1.5 GHz GSSI bow-tie antenna over a lossless half-space of permittivity $\varepsilon_r$ = 5, at distances from 0.1 to 0.57 m (H-plane, colors mark amplitude levels [20]).

The problem of characterizing the directional properties of ground penetration radar has been discussed in a number of classical works, e.g., see [13–15,32]. The simplest approach, in the far zone of harmonic radiation, is just the superposition of the Tx and Rx radiation patterns. The situation becomes more complicated for non-harmonic radiation, even in the model case of an infinitely long GPR line antenna [30] excited by a single current step [31]. This can be illustrated using the following set of color plots (Figure 5). The left frame (a) shows the spatial distribution of the pulsed radiation of an infinite linear vibrator placed on the ground at a certain moment ($ct = 1$ m). A similar spatio-temporal radiation pattern has a receiver antenna (b). The radar spatial resolution is determined by the product (c) showing a bright spot at the intersection of (a) and (b) patterns—cf. [15].

These analytical results are presented here in order to provide a rough qualitative picture of realistic GPR radiation characteristics and illustrate the problem of their quantitative estimates. First of all, the Engheta–Papas–Elachi model of an infinitely long line antenna [30] works only at distances small or similar, compared with the antenna length. Moreover, numerical calculations show that in time domain, the peak amplitude values tend to a far-field pattern with sharp edges at the critical total reflection angles $|\theta| = \eta$ quite gradually, at distances of hundreds of meters, unrealistic for most applications [31], cf. Figure 4b. Finally, Figure 5 highlights the complexity and ambiguity of the term "pulsed GPR radiation pattern", cf. [13–17].

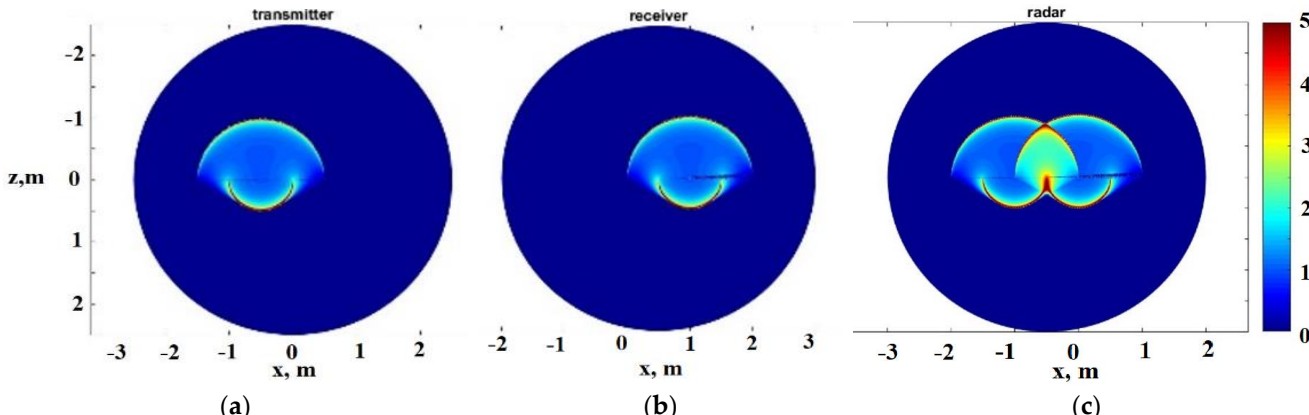

**Figure 5.** The spatio−temporal radiation pattern of an infinite line GPR, with 1 m separation between the antennas. (**a**) Transmitter pulse propagating into the ground ($n = 2$) for the distance $c\,t = 1$ m; (**b**) receiver directivity pattern; (**c**) Resulting GPR pattern.

The experimental evaluation of the radiation pattern of subsurface radar requires either the creation of an expensive measuring stand or the ingenious use of local objects as natural test objects [29]. The latter technique was used by participants of the Crimean expedition in 2022 in the spare intervals between archaeological works. In our measurements, natural test objects were at hand (power line wires, as shown in Figure 6, and non-metallic pipes of an ancient aqueduct were discovered during archaeological excavation, as shown in Figure 7).

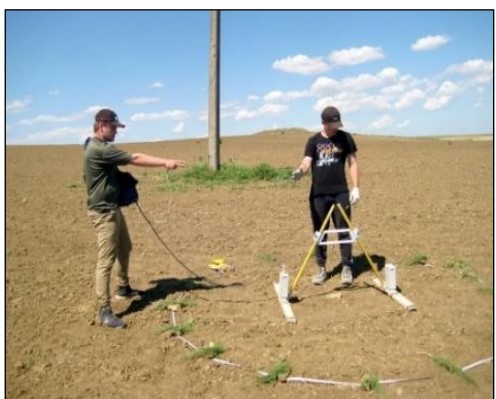

**Figure 6.** Measurements of GPR radiation pattern in the upper hemisphere under an electric power line.

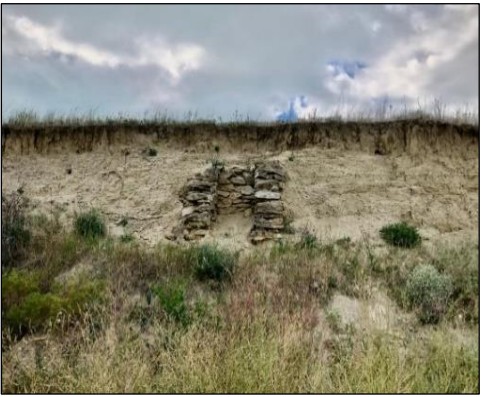

**Figure 7.** Ancient well and pipe outlets.

Before proceeding to the presentation of the experimental results, the following re-
marks should be made. We did not seek to express the characteristics of GPR emitting and
receiving systems in the exact formulas of the antenna theory [33]. Parameters such as gain
and directivity cannot be accurately measured in the field, and also do not have a strict
definition for monopulse GPR antennas due to the variability of the pulse waveform. In
addition, the resulting radiation pattern depends not only on the design and size of the Tx
and Rx antennas but also on the properties of the underlying surface [13,18–20], which may
change considerably along the measurement path. Nevertheless, a practical assessment
of the directivity characteristics is possible and very useful, whereas the calculation of the
antenna gain for an enhanced power GPR, such as Loza–V, is of no practical importance.

In order to confirm the need to take into account the radiation pattern of the GPR
antennas in the upper hemisphere, we provide an example of working with the Loza–V
GPR in a rural area. The task was to localize an underground cable along a high-voltage
power line. The primary B-scan across the route is shown in Figure 8a. First of all, the two
systems of hyperbolas strike the eye in the left and right parts of the figure, which could be
interpreted as an image of localized objects at a depth of 10 and 3 m, if it were not for the
excessively wide angular opening of the hyperbola wings, not leading to their focusing
by standard wave migration methods [34,35], as shown in Figure 8b. The soil dielectric
constant $\varepsilon \sim 4$ was assessed when using the standard CMP method [1] (other practical
methods for estimating the electrical parameters of the soil are discussed, for example, in
Refs. [7,9]). Visual control of the situation prompted the operator to perform the wave
migration procedure with a value $\varepsilon = 1$ corresponding to the dielectric constant of free
space, as shown in Figure 8c. As a result, we obtained an impressive picture of five wires of
a high-voltage power line at a height of 19–25 m, compared with Figure 8d, and a single
overhead cable at a height of about 6 m in the right corner of the picture. An image of the
desired underground cable is visible as a narrow hyperbola in the upper-right corner of
Figure 8a and as a faint spot in Figure 8b.

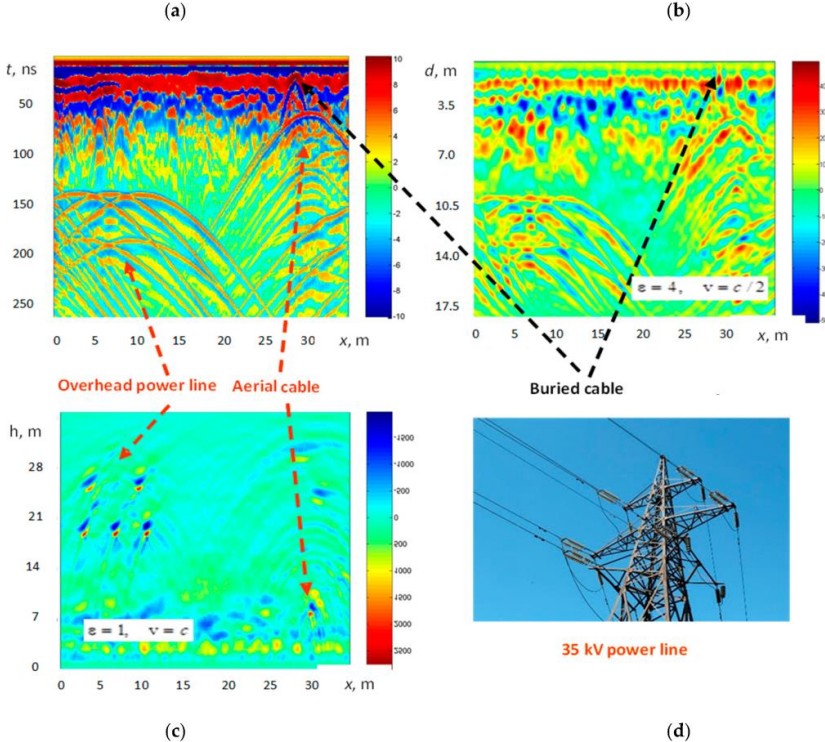

**Figure 8.** GPR search of a buried cable in a rural area. (**a**) Primary B–scan; (**b**) wave focusing with
soil dielectric constant $\varepsilon \sim 4$; (**c**) wave focusing with vacuum permittivity value $\varepsilon = 1$; (**d**) overhead
electric power line.

The use of natural linear objects to evaluate the Loza–V GPR directivity pattern in the upper and lower hemispheres is described below in Sections 3.1 and 3.2. Sections 3.3 and 3.4 are devoted to the phenomenon of the Patom crater, where we discuss the results of the GPR cross-section of this natural object and apply the solution of the 1D inverse problem to explain the peculiarities of the deep subsurface sounding.

## 3. Experimental Results

### 3.1. GPR Directivity Pattern in Upper Hemisphere

In the spring of 2020, scientists from the Institute of Archeology of the Crimea and the Institute of Archeology of the Russian Academy of Sciences identified one of the earliest Jewish necropolises in the vicinity of the modern city of Kerch. The site of the necropolis belonged to the community of Panticapaeum, mentioned by ancient authors and located at Cape Ak-Burun. It was here that the archeologists collected more than three dozen Jewish tombstones and fragments containing inscriptions and images (Figure 8a). The revealed names of the deceased members of the Jewish community add to our knowledge of the onomastics of the "Bosporan" civilization.

In the framework of the 2022 expedition, the territory of the necropolis was studied via Loza–V scanning, with 100 MHz antennas and a 5 kV transmitter. A dense accumulation of anomalies, interpreted as graves, was found in a relatively small area. Five previously unknown objects on the radar image were identified as crypts, and up to 40 tombstones with Jewish symbols were found on the site (Figure 9). Further study of the necropolis during the 2023 summer field campaign allowed the archeologists to trace the path of Jewish migration to the Bosporus in the 1st century BC and the first centuries of our era.

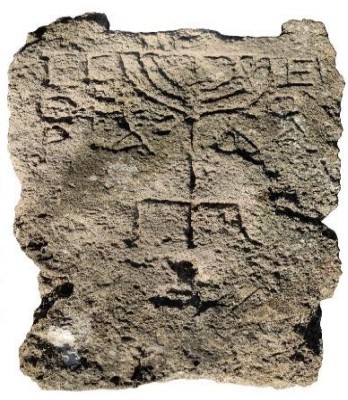
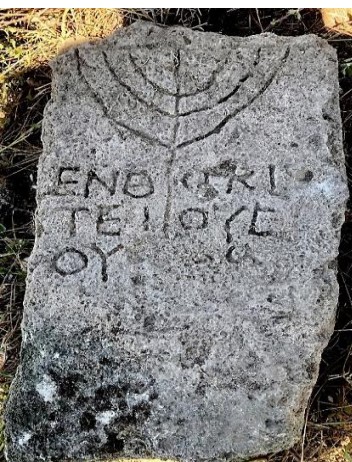

**Figure 9.** Fragments of Jewish tombstones.

The first purely radiophysical task in this expedition was to evaluate the GPR radiation patterns in the upper hemisphere. This was performed according to the following method: the Loza–V GPR antenna set was installed in the center of a circle divided into 30-degree sectors (Figure 6). In each of the positions, four measurements were taken, after which the antennas were rotated by 30 degrees, with the radar handle remaining above the center of the circle. The elongated scattering test object was a power line wire at a height of 15 m (one of the pillars is visible in Figure 6). The operator moved under the power line in a circle synchronously with the rotation of the antennas.

As a result of the GPR signal measurements, the horizontal radiation pattern of the 200 MHz dipole Loza–V antenna in the upper hemisphere (in air) under real field conditions (on wet ground with $\varepsilon_r \sim 8$) was obtained. From visual observation, it was understood that in the radargram, the subsurface reflections could be mixed with the echo of the power line above the GPR set at a height of about 15 m (round-trip delay of about 100 ns) (see Figure 6). In order to obtain a realistic radar image, clearly seen on the

radargram and B-scan (Figure 10a), it was converted using "radar velocity" $v_{radar} = 0.5$ $c = 15 \cdot 10^7$ m/s corresponding to $\varepsilon_r = 1$. The resulting radiation pattern, calculated as the angular dependence of the relative maximum of the radar pulse scattered by the power line cable, is shown in Figure 10b. The difference between the maximum and minimum of the directivity pattern is more than 40 dB (it is interesting to note that the radiation pattern has some asymmetry; the closer the receiver antenna is to the object, the greater the amplitude of the reflected signal). The obtained estimate of the radiation pattern is of practical importance when planning a GPR survey in urban conditions with unremovable objects of "aerial" reflection.

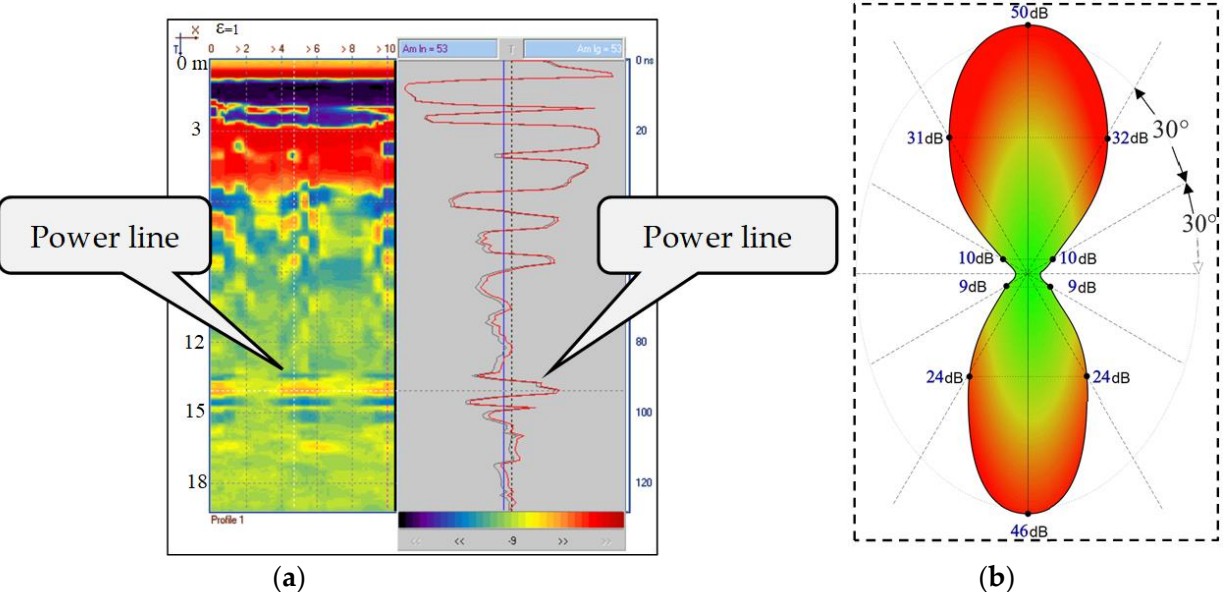

(**a**)                                          (**b**)

**Figure 10.** (**a**) GPR B–scan of the electric power line. Conversion of the radar delay time in the apparent "object depth" is performed with $\varepsilon_r = 1$ ($v_{radar} = 0.5\,c$); (**b**) Horizontal directivity pattern. Amplitude color scale is shown in the bottom of (a) plot (relative units).

### *3.2. GPR Directivity Pattern in Subsurface Medium*

The evaluation of the Loza–V GPR directivity pattern in the subsurface medium was performed during the Crimean summer expedition of 2022 at two archeological sites. One of them was found in the area of the Hebrew necropolis mentioned in Section 3.1. It was a crypt examined at the request of the Institute of Archeology of Crimea. The crypt was opened earlier by robbers and had no archeological value. However, it was useful for us as a reference for identifying the radar image of a known object and estimating the GPR radiation pattern. The radargram in Figure 11a shows both the measured radar delay time $t$ in nanoseconds (right scale) and the object depth $d$ in meters, converted using the estimated "radar velocity" $V_r = \frac{c}{2\sqrt{\varepsilon}} \approx 5 \cdot 10^7$ m/s (left scale). The measured depth at the top of the crypt (3.5 m) was taken into account in the interpretation of the radargram. The measured radiation pattern, as shown in Figure 11b, reflects the good focusing of the GPR signal and the high level of the reflected pulse, which is due to the high dielectric permittivity and low loss rate of the propagation medium.



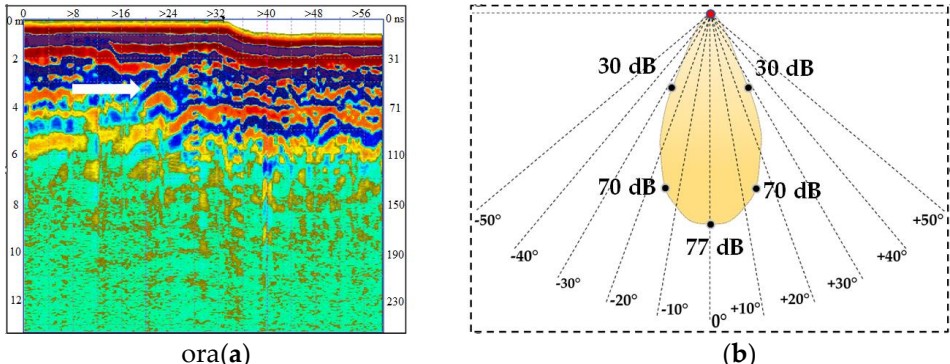

ora(**a**)                                                                                    (**b**)

**Figure 11.** (**a**) Subsurface radar B–scan of the ancient Jewish cemetery (Kerch area). The image of the ancient crypt top is marked with a white arrow. The same color scale as in Figure 10; (**b**) Measured directivity pattern of Loza–V GPR calculated for wet soil with $\varepsilon_r - 8$ (orange color intensity reflects field strength).

Another test object for GPR directivity measurements was found in the course of archeological work near the village of Geroevka (El Tiygen). In its vicinity, about 2.5 km from the village, there is an ancient well (Figure 12). According to archaeologists, this well and other objects unearthed as a result of landslides or during the construction of a gas pipeline (Figure 13a), are related to the ancient settlement of Nymphaeum.

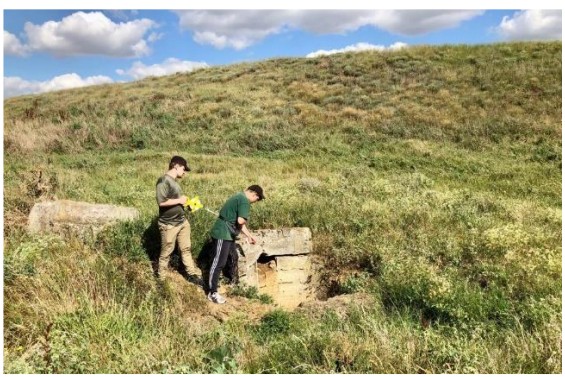

**Figure 12.** Ancient well.

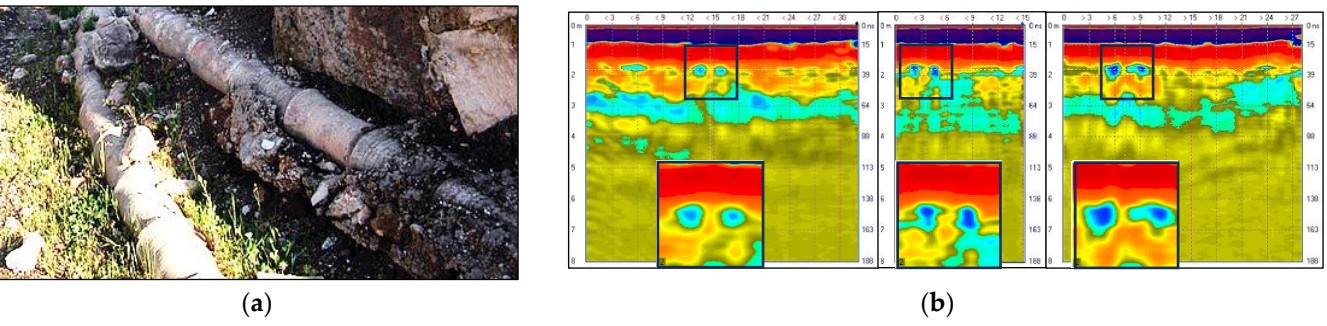

(**a**)                                                                                    (**b**)

**Figure 13.** The photo of antique water supply ceramic pipes (**a**). Three B–scans revealing cross-sections of two ceramic pipes, with enlarged square inserts (**b**). The same color scale as in Figures 10 and 11.

The objective of our GPR survey was to obtain, in a non-disruptive manner, information confirming the existence of an ancient underground water supply in the area between the identified objects. It was carried out, following the instructions of archaeologists, using Loza–V GPR, 200 MHz antennas, and a 5 kV transmitter (attenuated by 30 dB).

A structure similar to a cross-section of two parallel non-metallic pipes was identified in all the marked areas, as shown in Figure 13b and A1 in Figure 14a,b. The radar image can be interpreted as the two ceramic pipes of an ancient aqueduct. Apart from the pure archeological interest, they served as a convenient elongated test object for evaluating the GPR antenna directivity under typical field conditions. Figure 14c illustrates the practical method of directivity pattern assessment.

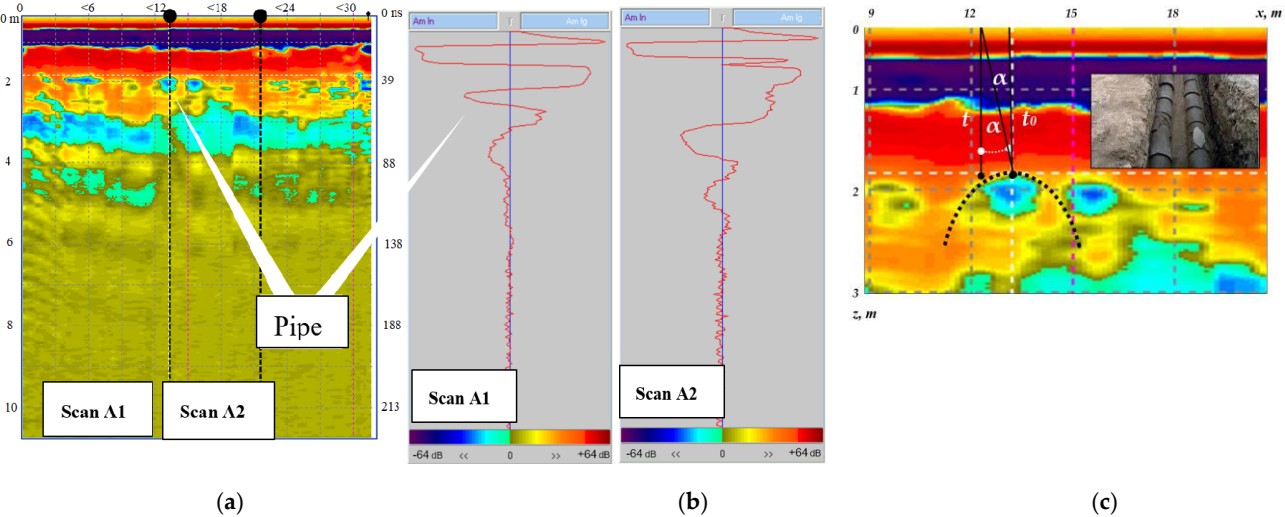

(a)  (b)  (c)

**Figure 14.** GPR B–scan with marked cross-sections A1 and A2 (**a**); two A–scans in the selected cross-sections, with logarithmic color scale. Scan A1 reveals the subsurface pipe at about 2 m depth (**b**); practical method of GPR directivity pattern assessment (**c**). The amplitude color scale is shown in the bottom of (**b**).

By moving horizontally at a distance $x$ from the top of the hyperbola (image of a compact object at a depth $z = h$), we increased the distance according to the law $\sqrt{h^2 + x^2} = h/\cos\alpha$. The amplitude of the reflected signal at the observation point $x$ is taken from the waveform in Figure 14b at the corresponding time delay $t = t_0/\cos\alpha$, where $t_0 = \frac{2h}{c}\sqrt{\varepsilon_r}$ (see Figure 14c). In this way, the effective radiation pattern $F(h, \alpha)$ of the transmit-receiving GPR antenna was built. Although it depends not only on the soil parameters but also on the depth and the scattering pattern of the elongated buried test object, these estimates are qualitatively true and very useful when working in typical field conditions. The results of the GPR radiation pattern measurements via radar pulse scattering by the buried pipe are shown in Figure 15.

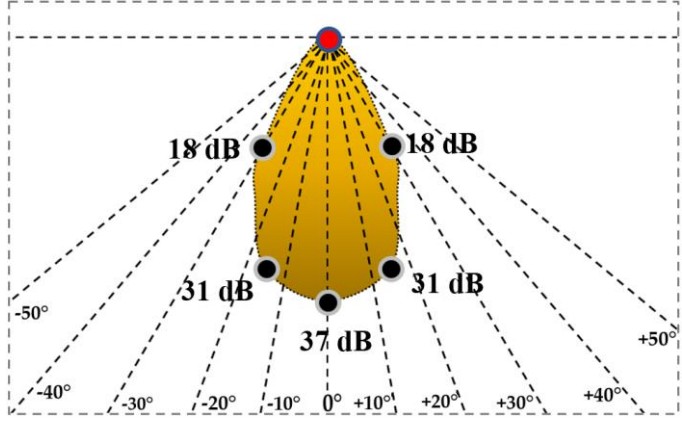

**Figure 15.** Loza–V GPR directivity estimated on the subsurface object shown in Figure 13a (yellow color intensity reflects field strength).

Another assessment of GPR antenna directivity was carried out during a survey of the Venetian settlement territory (Tikhaya Balka, XIV-XV centuries AD) at the order of the Archaeological Expedition of the State Hermitage Museum. As a test object, we used the foundation of the destroyed fortress wall at a depth of 1.5 m. This experiment gave a larger width of the main lobe—about 45 degrees at the level of 0.7 (see Figure 16). This difference might be due to the lower refractive index of dry sandy soil.

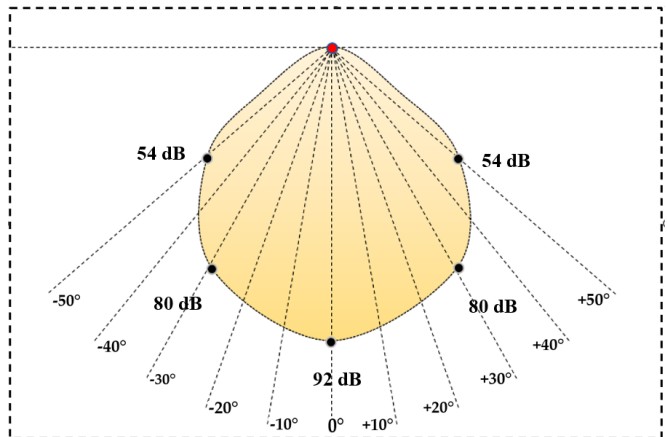

**Figure 16.** Loza–V GPR directivity on dry sand ground (orange color intensity reflects field strength).

In both cases, one can notice a rather smooth directivity pattern, which seemingly contradicts the classical Engetta–Papas–Elachi results [30] predicting "horns" of the GPR antenna radiation pattern at the angle of total internal reflection, as shown in Figure 4a. However, as thorough analysis, laboratory experiment, and numerical calculation confirm, realistic resistively loaded GPR antennas have smooth radiation patterns [19,20,32].

### 3.3. Deep Radar Probing of Patomsky Crater

In what follows, we discuss the experimental material obtained during the research mission organized in September 2022 by the Russian Geographic Society, the newspaper "Komsomolskaya Pravda" and the TV channel "Russia1". The object of the study was the Patomsky crater, an unusual geological structure in the Siberian taiga discovered in the Bodaibo district of the Irkutsk region [26]. This rare object (a stone ring 80 m in diameter, with a central cone composed of large fragments of limestone, as shown in Figure 17) attracted the attention of geo-scientists. The organizers invited experienced geophysicists V. L Sulyandikov and F. P. Morozov for the scientific support of the project.

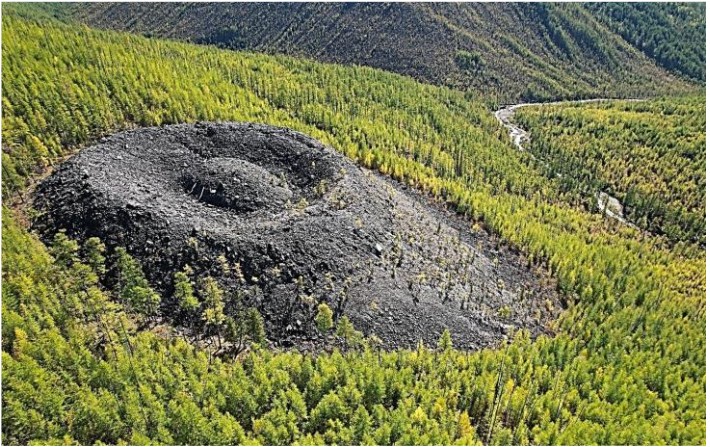

**Figure 17.** Patomsky crater—general view.

The origin of the mysterious Patomsky crater, located in the north of the Irkutsk region, attracted the attention of many researchers. V.V. Kolpakov, who discovered the Patomsky crater in 1949 during geological survey work, put forward a hypothesis about its formation as a result of a meteorite fall [26]. The famous volcanologist S.V. Obruchev argued with this hypothesis, leaning toward the volcanic origin of the crater. This point of view was seriously supported by geological and dendrological studies [27]. Nevertheless, some experts defend the meteorite hypothesis, assuming the presence of a foreign body in the Patomsky crater at a depth of 180–200 m [28].

Our GPR cross-sections (Figure 18) performed by F. P. Morozov confirm the most probable volcanic model [27]. The survey was carried out from the Earth's surface using a JSC VNIISMI low-frequency Loza–N DPR [9,26] equipped with a 21 kV pulse transmitter and antennas with a central frequency of 25 MHz (antenna length of 6 m). Unlike high-frequency 200–400 MHz GPR [36,37] designed for probing the first subsurface meters, the time scale for this measurement was set to 4096 ns (maximum depth of about 200–250 m). Similarly to Figure 10, both radar delay time (right scale) and object depth (left scale) are shown in Figure 19b and below, with an estimated "radar velocity" of 5.5 cm/ns.

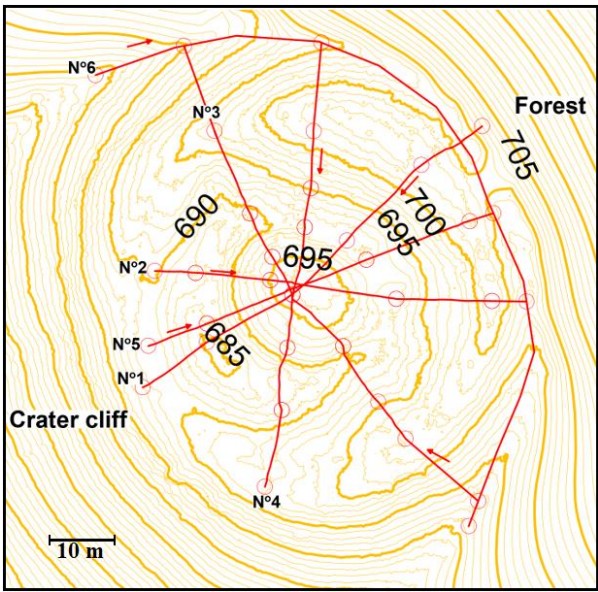

**Figure 18.** GPR survey paths over the Patomsky crater (numbered red lines, passed along the arrows). Isohypses with heights are marked in meters (yellow lines).

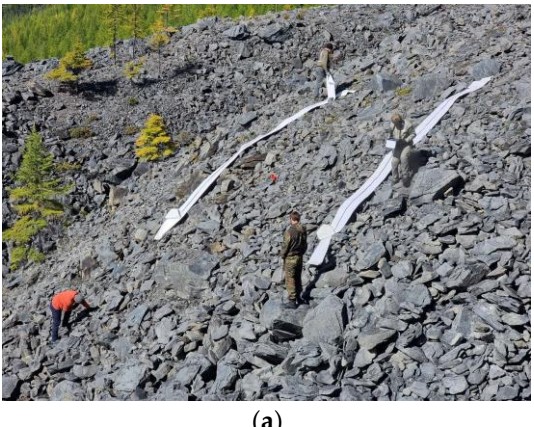

(**a**)

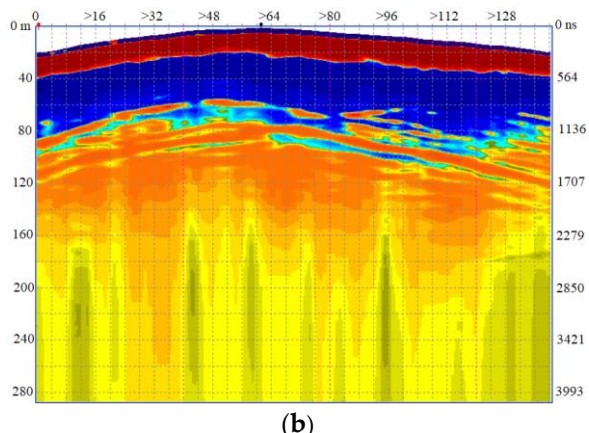

(**b**)

**Figure 19.** (**a**) GPR probing of the crater outer ring; (**b**) Radar B–scan No 6. Here and below a conditional amplitude scale is used.

GPR B–scan No 6 recorded along the outer ring of the crater (Figure 19a) reflects the geological structure outside the crater cone. According to the interpretation of experienced geologists, it is composed of the following elements. Horizon 0–60 m is represented by limestone with a homogeneous undisturbed structure. Deeper than 60 m, undisturbed rocks are recorded, markedly differing in physical properties from the overlying horizon. The main result of the GPR data analysis along this profile is that there are no vertical anomalous structures outside the crater.

The next B–scan, No 3, was recorded along the latitudinal diameter of the crater (Figure 20a). Here, the geological structure is represented by the following features: horizon 0–100 m is filled with destroyed limestone; undisturbed rocks lie deeper. The most interesting anomalous structure of vertical development is recorded on the central hill of the crater. Two more similar smaller-scale anomalies are noted in the ring moat on the northern and southern sides of the crater—see Figure 20b.

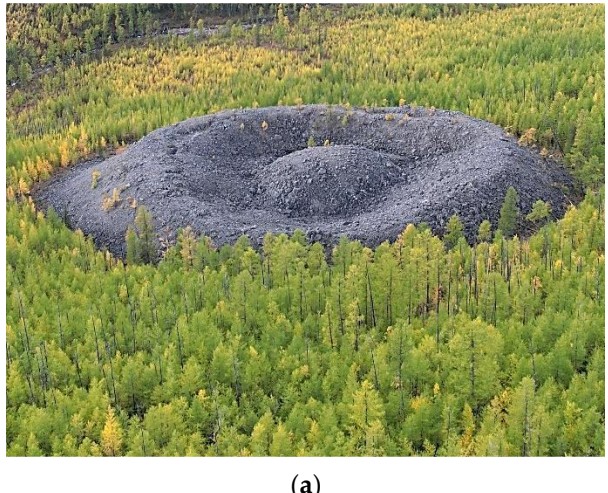
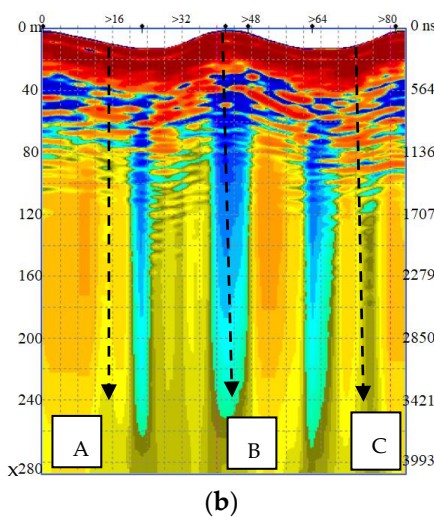

(**a**)　　　　　　　　　　　　　　　　　　　　(**b**)

**Figure 20.** GPR probing across the latitudinal diameter of the crater. (**a**) Aerial photo; (**b**) GPR B–scan No 3 with marked A–scans (A, B, C ) at 12, 40 and 72 meters of the profile.

A series of horizontal cross-sections of the obtained GPR data (Figure 21a) made it possible to construct an impressive 3D electrodynamic model of the Patomsky crater (Figure 21b). The depths of the selected cross-sections were estimated using a hypothetical radar velocity of $v_r = 5.5$ cm/ns.

Of special interest are the deepest sections—from 180 to 280 m. All of them clearly show the contour of the central vertical anomalous structure, located under the central hill of the Patomsky crater. At depths of less than 200 m, anomalous structures of a smaller scale appear. It is noteworthy to mention that the well-distinguishable ring structures around the main anomaly are registered in all sections. Such ring structures (concentric zones of compression and expansion) are typical for the areas of explosive impact on the rock.

Additional information about the nature of the Patomsky crater is provided by the analysis of the reflected signal waveforms—see Figure 22. The signals (A) and (C), measured above the rock mass, are characterized by the amplitude deviation towards negative values (−53 and −51 dB, respectively). Such properties of the reflected signal, according to the results of theoretical analysis, indicate low values of the propagation medium permittivity [11]. In practice, this means that the probing signal propagates in a high-resistance medium (in this case, monolithic limestone).

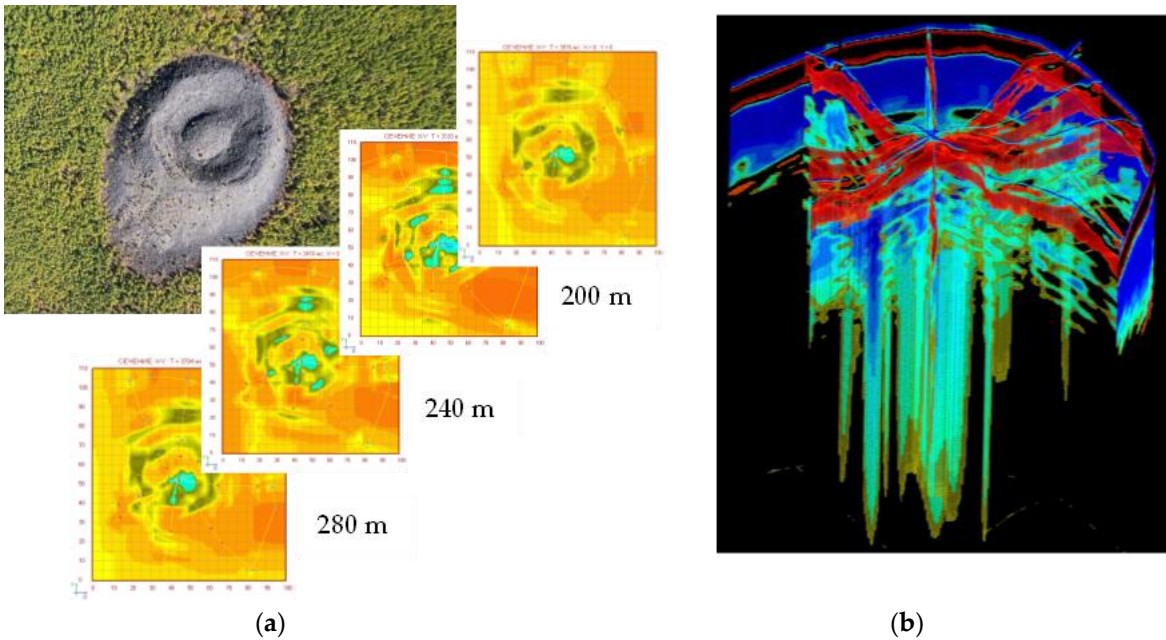

**Figure 21.** Loza–N GPR qualitative image of the Patomsky crater. (**a**) Horizontal sections; (**b**) 3D electrodynamic model.

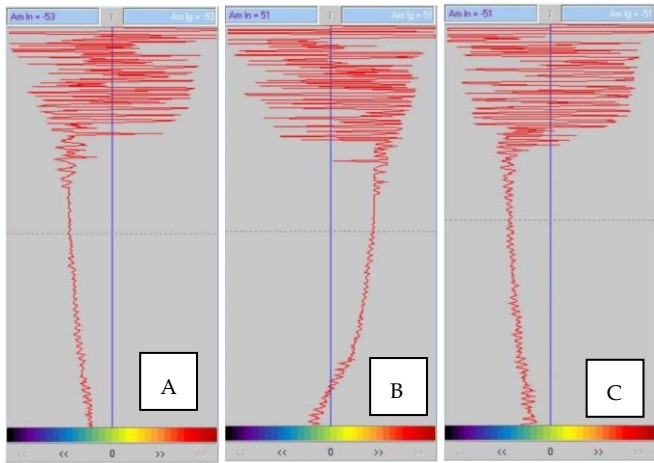

**Figure 22.** Three GPR signal waveforms from B–scan shown in Figure 20b: $x = 12, 40, 72$ m (**A–C**) and the amplitude color scale.

Waveform (B) shows a significant deviation in the signal amplitude in the region of positive values.

The deviations in the reflected signal amplitude indicate that the EM wave propagates in a medium with high values of permittivity or conductivity [38–40]. Such changes in the properties in a limited vertical portion of the limestone mass can only be associated with a local influx of moisture, the appearance of which can substantially increase the dielectric constant. A local fracture, which can arise due to tectonic processes or from a local impact, can serve as a collector for moisture ingress into the rock mass.

The characteristic shape of the Patomsky crater surface, which consists of a ring structure with a bulk cone, an annular moat, and a central hill, can also, with some probability, indicate that the crater originally arose due to the impact of a meteorite, which would support the original hypothesis of A. M. Portnov [28]. The existence of such a local channel of limited depth, which ensures the transport of surface waters in the cold Siberian climate, would lead to the second mechanism—ice diapirus (frost heaving). The processes

of frost heaving every winter straighten and restore the shape of the Patomsky crater with a bulk cone, annular ditch, and central hill. The latter version also seems quite probable. The only indisputable fact is the existence of a water-saturated channel in a homogeneous limestone mass under the central hill of the Patomsky crater, which was convincingly confirmed by the georadar.

*3.4. Qualitative Solution of Inverse Problem*

Many analytical and numerical approaches have been developed to solve the problem of GPR pulse propagation in a smoothly layered subsurface medium—e.g., see [41]. For a qualitative analysis of the aforementioned GPR survey results, we used our approximate solution of the one-dimensional inverse problem of electromagnetic sounding [42,43]—a time-domain version of the classical coupled wave method [44,45]. It provides a general picture of the reflected pulse formation on the subsurface medium gradients, consistent with numerous experimental results, and yields a closed-form solution of the simplified inverse problem. We represent the electromagnetic signal received by the GPR antenna as $E(0, s) = f(s) + g(s)$. Here, $s = ct$ is the normalized propagation time of the probing pulse ($c$ is the speed of light), $f(s)$ is the signal waveform, and the integral

$$g(s) = -\frac{1}{4} \int_0^\infty \frac{\varepsilon'(z)}{\varepsilon(z)} f\left( s - 2 \int_0^z \sqrt{\varepsilon(z)} d\zeta \right) dz \qquad (1)$$

describes the cumulative effect of partial reflections of the incident wave from dielectric permittivity gradients in the subsurface medium. This approximation, obtained using the coupled wave method [31], makes it possible to find an explicit solution $\varepsilon(z)$ to the inverse problem. Its parametric representation has the form

$$
\begin{aligned}
\varepsilon[z(s)] &= \varepsilon_0 \exp\left( -4 \int_0^s Q(s) ds \right) \\
z(s) &= \frac{1}{2\sqrt{\varepsilon_0}} \int_0^s \exp\left( 2 \int_0^s Q(r) dr \right) ds
\end{aligned}
\qquad (2)
$$

where the function $Q(s) = -\frac{\varepsilon'[z(s)]}{8\varepsilon^{3/2}[z(s)]}$ is determined by the inverse transform of the ratio

$$Q(s) = \frac{1}{2\pi} \int_{-\infty+i\delta}^{\infty+i\delta} \widetilde{Q}(k) e^{-iks} dk, \quad \widetilde{Q}(k) = \frac{\widetilde{g}(k)}{\widetilde{f}(k)} \qquad (3)$$

of the incident and reflected pulse Fourier transforms: $\widetilde{f}(k) = \int_0^\infty f(s) e^{iks} ds$ and $\widetilde{g}(k) = \int_0^\infty g(s) e^{iks} ds$. Absorption can be taken into account by introducing the complex permittivity, $\widetilde{\varepsilon} = \varepsilon + 4\pi i \frac{\sigma}{\omega}$, which leads to a significant complication of the analysis [31]. Under the condition $4\pi\sigma << \omega$, the perturbation method can be used. The calculations are simplified and lead to a slight modification of the main formula, Formula (1):

$$g(s) = -\frac{1}{4} \int_0^\infty \frac{\varepsilon'(z)}{\varepsilon(z)} \exp\left( -\frac{4\pi}{c} \int_0^z \frac{\sigma(\zeta)}{\sqrt{\varepsilon(\zeta)}} d\zeta \right) f\left( s - 2 \int_0^z \sqrt{\varepsilon(\zeta)} d\zeta \right) dz \qquad (4)$$

Unfortunately, one measurement of the reflected signal waveform is not enough to determine the two unknown functions: $\varepsilon(z)$ and $\sigma(z)$. To determine them, one can use an a priori electrodynamic model of the subsurface medium (for example, by assuming a constant loss angle: $q = \arctan\frac{4\pi\sigma(z)}{c\varepsilon(z)} = Const$) or perform two measurements of the

reflected signal with the antennas having different frequency responses [42]. In the first case, we obtain an integral equation

$$g(s) = -\frac{1}{8}\int_0^s \frac{\varepsilon'[z(r)]}{\varepsilon^{3/2}[z(r)]} \exp\left(-\frac{q}{2}r\right) f(s-r)\, dr = \int_0^s P(r)f(s-r)\, dr, \tag{5}$$

with $P(s) = \frac{1}{2\pi}\int_{-\infty+i\delta}^{\infty+i\delta} \frac{\widetilde{g}(k)}{\widetilde{f}(k)} e^{-iks}\, dk = -\frac{\varepsilon'[z(s)]}{8\varepsilon^{3/2}[z(s)]} \exp\left[-q\int_0^z \sqrt{\varepsilon(\zeta)}\, d\zeta\right]$ and some generalization of the parametric solution (2)–(3):

$$
\begin{aligned}
\varepsilon[z(s)] &= \varepsilon_0 \int_0^s \exp\left(-4\int_0^s P(r)\exp(\tfrac{q}{2}r)dr\right) \\
z(s) &= \frac{1}{2\sqrt{\varepsilon_0}}\int_0^s \exp\left(2\int_0^s P(r)\exp(\tfrac{q}{2}r)dr\right) ds
\end{aligned}
, \tag{6}
$$

The second approach requires some complications of the experimental technique.

In this work, to obtain a qualitative estimate, we neglect ohmic absorption, which is apparently not very significant under experimental conditions, and consider the level of the reflected signal as a measure of the vertical gradients of the subsurface medium permittivity. For an asymptotic estimate, we transform Equation (5) via integration by parts, defining $f(0) = f(\infty) = 0$, $f(s) = \frac{d}{ds}h(s)$, where the function $h(s)$ has a single maximum at $s = l$. Let, for example, $h(s) = 1 - \cos(2\pi e^{-as})$, where $l = \frac{\log 2}{a}$ is the characteristic pulse length. For a short pulse, integral (5) is determined using a narrow neighborhood of the function $h(s)$ maximum. By removing slowly varying functions from under the integral sign (5), we obtain the following:

$$
\begin{aligned}
g(s) &= -\frac{1}{8}\int_0^s \frac{d}{ds}\left[\frac{\varepsilon'[z(r)]}{\varepsilon^{3/2}[z(r)]} \exp\left(-\frac{q}{2}r\right)\right] h(s-r)\, dr \\
&\approx -\frac{1}{16}\left[\varepsilon^{-1/2}\left(\frac{\varepsilon'}{\varepsilon^{3/2}}\right)' e^{-\frac{q}{2}r}\right] \int_0^s h(s-r)\, dr
\end{aligned}
. \tag{7}
$$

The integral $H(s) = \int_0^s h(s-r)\, dr = s + \frac{1}{a}[ci(2\pi e^{-as}) - ci(2\pi)]$ for small values of $s$ can be approximated using a linear function $H(s) \approx 2s - 0.86\,l$, which rapidly tends to the constant $H(s) \approx 3.5\,l$ depending on the antenna half-length $l$ (see Figure 23).

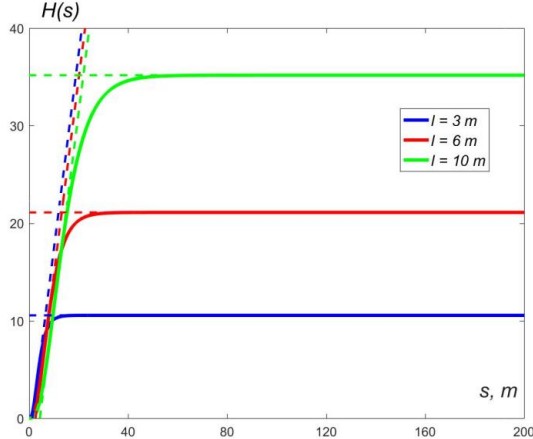

**Figure 23.** Model source function $H(s)$ for $l = 3, 6, 10$ m. Explanation of colors is given in the box; the dashed lines mark the common tangent.

Following the method developed in our works [42,43], we consider Formula (7) as a differential equation for determining the ground permittivity profile $\varepsilon(z)$. Indeed, for a given model of the GPR pulse $f(s)$ and the measured waveform of the reflected signal waveform $g(s)$ in the differential equation $\left[\varepsilon^{-1/2}(z)\left(\frac{\varepsilon'(z)}{\varepsilon^{3/2}(z)}\right)' - q\frac{\varepsilon'(z)}{\varepsilon^{3/2}(z)}\right] = 4\,\Phi(s)$, the right-hand side $\Phi(s) = -4\frac{g(s)}{H(s)}$ is a known function that can be integrated by substitution $\varepsilon(z) = \varepsilon_0 \exp V(s)$, $s = 2\int_0^z \sqrt{\varepsilon(r)}dr$: $\frac{\varepsilon'(z)}{\varepsilon^{3/2}(z)} = 2\,\dot{V}(s)$, $\left(\frac{\varepsilon'(z)}{\varepsilon^{3/2}(z)}\right)' = 4\,\varepsilon^{1/2}\ddot{V}(s)$; $\ddot{V}(s) - \frac{q}{2}\dot{V}(s) = \Phi(s)$. Taking into account the initial condition $\varepsilon(0) = \varepsilon_0$ its explicit solution is given by the integral

$$V(s) = \frac{2}{q}\left\{\int_0^\infty \exp\left(-\frac{q}{2}r\right)[\Phi(r) - \Phi(r+s)]\,dr - \int_0^s \Phi(r)\,dr\right\} \tag{8}$$

Equation (9) is simplified in the absence of losses ($q = 0$):

$$\varepsilon(z) = \varepsilon_0 \exp V(s),\, V(s) \approx \int_0^\infty [\Phi(r+s) - \Phi(r)]\,r\,dr = \int_0^\infty \Phi(r)\min(r,s)dr \tag{9}$$

In combination with the integral $z = Z(s) = \frac{1}{2\sqrt{\varepsilon_0}}\int_0^s \exp\left[-\frac{1}{2}V(r)\right]dr$, Equation (9) yields an explicit solution to the subsurface sounding problem. Three issues remain as follows:

(1) There is no registration of the emitted pulse amplitude in the current GPR models (some approaches to the problem are outlined in [46]);
(2) Since drilling near this rare natural object is excluded, the numerical value of the permittivity at the depth of far reflections can be estimated only from model calculations (we tried several likely values for different moisture saturation).

Moreover, our 1D model does not describe the divergence of the probing signal in the subsurface medium, which leads to some errors in the return pulse amplitude. If necessary, this effect can be taken into account by introducing an appropriate divergence factor.

With these considerations, an approximate solution of the 1D inverse problem was used to process the GPR sounding data of the Patomsky crater, as performed by F. P. Morozov, as a part of the Komsomolskaya Pravda expedition. Three A-scans were selected in Figure 24a: in the center of the stone "dome" (A 38) and at two points on the annular shaft—A 21 and A 48. For convenience, the color amplitude scale used throughout the paper is repeated in Figure 24b. Attention was paid to the difference in the sign of deep reflections when sounding the central dome and its periphery—see Figure 22. To construct the dielectric permittivity profile, taking into account the above considerations, our analytical solution was written in a form convenient for practical assessments, with explicitly indicated hypothetical permittivity values at the reflection depths of the initial and tail parts of the GPR pulse $\varepsilon_0 = \varepsilon(0)$ and $\varepsilon_1 = \varepsilon(z_1)$:

$$\varepsilon = \varepsilon[Z(s)] = \varepsilon_0\left(\frac{\varepsilon_1}{\varepsilon_0}\right)^{R(s)},\quad R(s) = \int_0^\infty \frac{g(r)}{H(r)}\min(r,s)\,dr \Big/ \int_0^\infty \frac{g(r)}{H(r)}\,r\,dr\,. \tag{10}$$

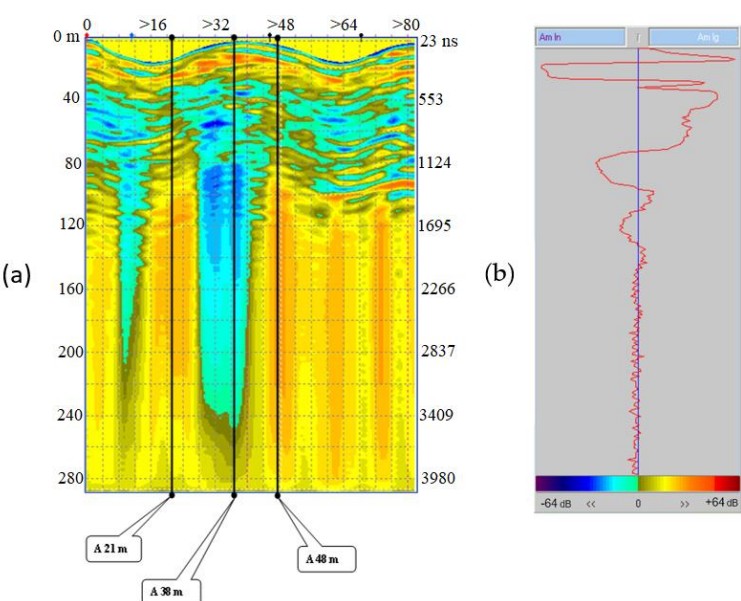

**Figure 24.** (**a**) Radargram of deep subsurface probing of the Patomsky crater. B-scan with marked profiles: A 21 m, A 38 m, and A 48 m; (**b**) Logarithmic amplitude color scale is shown in the bottom of the linear plot.

Figure 25 shows plots of the soil dielectric permittivity reconstructed using Equation (10) from the initial value consistent with geological data to the deep ground values, corresponding to the hypothesis of meltwater saturation of the crater, for three hypothetical values of $\varepsilon_1 = \varepsilon(z_1) = 8,\ 12,\ 16$.

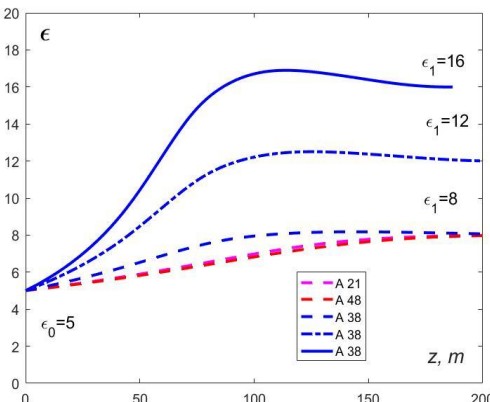

**Figure 25.** Depth permittivity profile in the center (A 38 m) and on the periphery of the crater (A 21 and A 48 m) reconstructed from the GPR data.

The maximum dielectric permittivity at the depths of about 100 m may indicate increased moisture saturation of the soil filling the well.

These estimates are rather speculative and preliminary. A detailed study of the Patomsky crater structure requires additional georadar and magnetometric sounding. The only conclusion that can be drawn from the DPR measurements is the absence of a compact foreign body in the crater, which would result in a noticeable backscattering.

## 4. Conclusions

The results of our Crimean campaign have cultural, historical, and methodological value. The usefulness of georadar surveys in archeological research can hardly be overestimated; the number of new artefacts found during a short GPR inspection of an ancient

Jewish cemetery in the vicinity of Kerch City is comparable with the total volume of previous findings. Archeological work near the village of Geroevka not only helped to outline the territory of a rare ancient water pipeline but may also guide the direction of further construction works. The objects found in the territory of the ancient Venetian settlement of the XIV-XV centuries in Tikhaya Balka will also contribute to the material history of the Middle Ages.

From a technical point of view, the Crimean expedition provided a good chance to estimate the GPR antenna radiation patterns in different materials (air, dry, or wet soils) and emphasized the necessity to organize quantitative measurements in typical media with controlled electrophysical parameters.

Our Siberian mission must be evaluated from different points of view. First of all, it was the first attempt at the objective electrophysical inspection of this rare natural object. The technical parameters of the enhanced-power GPR allowed us to obtain a radar image of the Patomsky crater up at a depth of about 200 m. The results of the deep georadar sounding did not support the hypothesis of a massive foreign body present in the crater, which makes the less exotic volcanic origin of the Patomsky phenomenon more probable. Other non-destructive techniques, such as multi-frequency radar probing combined with magnetic measurements, can provide new information on the origin and structure of the crater. In this regard, as well as to popularize the tourist route and organize the protection of this unique natural site, the "Komsomolskaya Pravda" expedition can play a good role.

**Author Contributions:** Conceptualization, P.M. and A.P.; methodology, P.M.; software, A.P. and M.L.; investigation, F.M., M.L. and L.B.; formal analysis, A.P.; resources, P.M.; data curation, F.M., M.L. and L.B.; writing and funding acquisition, P.M. and A.P.; English editing, M.L. All authors have read and agreed to the published version of the manuscript.

**Funding:** This research was funded by the Russian Science Foundation in the framework of the research project "Deep Penetration Radar: Theory, Methods, Experiment", grant № 22-1200083. Scientific management and technical support have been provided by IZMIRAN and JSC Company VNIISMI.

**Data Availability Statement:** The data presented in this study are available upon request from IZMIRAN and JSC VNIISMI.

**Acknowledgments:** The first part of the study was conducted with legal support from the Institute of Archeology of Crimea of the Russian Academy of Sciences, the State Hermitage Museum, and the Research Center for the History and Archeology of Crimea. The Siberian expedition was organized and supported by the editors of the newspaper "Komsomolskaya Pravda", and the scientific equipment was provided by VNIISMI and JSC Lenzoloto.

**Conflicts of Interest:** The Authors Pavel Morozov and Fedor Morozov were partly employed by the JSC Company VNIISMI, with no conflict of interest. The remaining authors declare that the research was conducted in the absence of any commercial or financial relationships that could be construed as a potential conflict of interest. The funders had no role in the design of the study; in the collection, analyses, or interpretation of data; in the writing of the manuscript; or in the decision to publish the results.

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
