# Peer review of "Characterization of Antenna Radiation Pattern and Penetration Depth in Ground Penetrating Radar Field Missions"

_remotesensing, doi:10.3390/rs15235452_

Round 1

Reviewer 1 Report

Comments and Suggestions for Authors

Article

Characterization of antenna radiation pattern and penetration depth in GPR field missions

General Comments:

1)      There is no clear results in the “Abstract”.

2)      The “Abstract” need to be detailed.

3)      The experimental part of the manuscript must be clear, and not presented in the “Introduction” section.

4)      The lines “121-130” must be transformed to the “Introduction” section.

5)      For the applied case study, as showed in figure (8), the GPR survey must be run with the two options (option 1: the normal direction of antenna, and option 2: the radiation pattern), to compare the collected GPR records.

6)      Line 475: “As it was stated in the Introduction”, no need for this statement.

Comments on the Quality of English Language

Author Response

Thank you for your valuable comments and suggestions.

1)      There is no clear results in the “Abstract”.

2)      The “Abstract” need to be detailed.

Reply: Abstract has been done more precise.

3)      The experimental part of the manuscript must be clear, and not presented in the “Introduction” section.

Reply: Only a brief record of the experimental work remained.

4)      The lines “121-130” must be transformed to the “Introduction” section.

Reply: We considered an explanation of the ambiguity of the radiation pattern term in time domain to be pertinent in the “Methods” section.

5)      For the applied case study, as showed in figure (8), the GPR survey must be run with the two options (option 1: the normal direction of antenna, and option 2: the radiation pattern), to compare the collected GPR records.

Reply: Unfortunately, we have no possibility to repeat measurements on this site.

6)      Line 475: “As it was stated in the Introduction”, no need for this statement.

Reply: Agree, removed.

(The changes are marked in blue color in the pdf version).

Reviewer 2 Report

Comments and Suggestions for Authors

An interesting paper to read. The description of the archeological and crater sites and display of the accompanying GPR data was very well done. However, the above-surface directivity calculation was not clear and the subsurface directivity calculations weren't theoretically accurate because they involved reflecting targets with unknown scattering properties. With some editing to fix these issues and several other minor ones (see attached word document edits and comments) it will be a very solid paper.

Comments on the Quality of English Language

The English was in general very acceptable. Several grammatical suggestions are noted in the attached word document.

Author Response

Thank you for your kind words and useful suggestions.

Reviewer:

An interesting paper to read. The description of the archeological and crater sites and display of the accompanying GPR data was very well done. However, the above-surface directivity calculation was not clear and the subsurface directivity calculations weren't theoretically accurate because they involved reflecting targets with unknown scattering properties. With some editing to fix these issues and several other minor ones (see attached word document edits and comments) it will be a very solid paper.

Reply:

We have added an additional explanation of the directivity evaluation in the upper hemisphere and rough qualitative estimation of the subsurface angular pattern. All minor errors are hopefully corrected (marked in green color in the pdf version).

Reviewer 3 Report

Comments and Suggestions for Authors

Dear Authors,

I recently had the opportunity to delve into your manuscript titled "Characterization of Antenna Radiation Pattern and Penetration Depth in GPR Field Missions," and I must say, it provides a commendable contribution to the field of ground-penetrating radar (GPR). You skillfully address the crucial aspects of predicting device resolution and probing depth, essential considerations when planning GPR missions.

The article particularly shines in its discussion of the methods and results pertaining to the assessment of the Loza-V and Loza-N radars. The inclusion of data obtained during cultural expedition works adds a practical dimension to the study, making it highly relevant to real-world applications in geology, archaeology, and civil engineering.

One notable strength of the article is its emphasis on the variability of the GPR antenna radiation pattern across different materials such as air, dry soils, and wet soils. The confirmation of this variability underscores the importance of quantitative measurements with controlled electrophysical parameters. This not only adds depth to the study but also serves as a valuable guideline for practitioners in the field, enhancing the applicability of the research findings.

As a suggestion to improve the manuscript is the literature review. A more comprehensive literature review would not only strengthen the theoretical framework but also provide readers with a better contextual understanding of the study's significance within the broader academic landscape. Please see some examples: 1. https://www.mdpi.com/2072-4292/15/13/3423; 2. https://www.mdpi.com/2072-4292/14/15/3659; 3. https://www.redalyc.org/journal/465/46574178004/html/; 4. https://www.flipkart.com/understanding-buried-anomalies-practical-guide-gpr/p/itm730218e9da8da

Furthermore, the recommendation for an English mother tongue expert in the field to review the article is another suggestion. Clear and precise communication is paramount in scientific writing, and having an expert review the language ensures that the nuances of the research are accurately conveyed to a global audience.

In conclusion, "Characterization of Antenna Radiation Pattern and Penetration Depth in GPR Field Missions" stands out as a valuable contribution to the GPR literature. Its meticulous approach to assessing key characteristics of GPR devices and the practical insights gained from field missions make it a must-read for researchers, professionals, and enthusiasts in the field. With the suggested improvements in the literature review and language review, this article has the potential to become a benchmark reference in the domain of GPR research.

Comments on the Quality of English Language

Dear Authors,

Ensuring the involvement of an English mother tongue expert in the field to review the article is a critical step towards enhancing the clarity and precision of the scientific communication. This recommendation is rooted in the understanding that the effectiveness of scholarly work goes beyond the accuracy of data and extends to how well the findings are articulated and understood by a diverse global audience.

Firstly, scientific writing demands a level of linguistic proficiency that goes beyond basic grammatical correctness. It requires an intricate understanding of the nuances, idioms, and conventions specific to the English language, particularly in the context of the field of study. An English mother tongue expert possesses an innate grasp of these linguistic intricacies, ensuring that the manuscript is not only grammatically sound but also resonates with the linguistic expectations of the scholarly community.

Moreover, the scientific community is diverse, with researchers and academics from various linguistic backgrounds contributing to and consuming the body of knowledge. By having an English mother tongue expert review the article, the authors take a proactive step in bridging potential language gaps. This is crucial for effective knowledge dissemination, as it ensures that the research is accessible and comprehensible to a broad audience, thereby maximizing its impact and relevance.

In the context of this specific article on GPR, where technical terms and specialized language are prevalent, precision in communication becomes even more vital. An English mother tongue expert with expertise in the field can refine the language to accurately convey complex concepts, preventing misinterpretations and fostering a clearer understanding of the research outcomes.

In essence, the involvement of an English mother tongue expert is not merely a formality but a strategic decision to elevate the quality of scientific discourse. It aligns with the ethos of robust and transparent communication, ultimately benefiting both the authors and the readership by ensuring that the intellectual contributions of the research are effectively communicated on a global scale.

Author Response

Dear Reviewer,

Many thanks for your kind words and useful suggestions.

Thank you for the relevant references. We incorporate them and some other papers in the reference list (Refs. [ 6-7, 35-37]).

Concerning your advice to involve a native English speaking expert in the field to review the article: we asked a prominent scientist, author of a classical treatise on X-ray and EUV optics Professor D.T. Attwood (Department of Electrical Engineering and Computer Sciences at the University of California, Berkeley): to read the article and give some comments. Here is a part of his letter:

Alex: The paper is so interesting. The combination of first-class physical science, anthropology, and earth science is wonderful. You and your group are very good and very fortunate to have this confluence of interesting work. I am enjoying it very much. And it is very well written for content, organization, and English. Dave ([email protected]).